# Targeting Proteasomes and the MHC Class I Antigen Presentation Machinery to Treat Cancer, Infections and Age-Related Diseases

**DOI:** 10.3390/cancers15235632

**Published:** 2023-11-29

**Authors:** Priyanka S. Rana, James J. Ignatz-Hoover, James J. Driscoll

**Affiliations:** 1Case Comprehensive Cancer Center, School of Medicine, Case Western Reserve University, Cleveland, OH 44106, USA; p.rana@case.edu (P.S.R.); james.ignatz-hoover2@uhhospitals.org (J.J.I.-H.); 2Division of Hematology & Oncology, Department of Medicine, Case Western Reserve University, Cleveland, OH 44106, USA; 3Adult Hematologic Malignancies & Stem Cell Transplant Section, Seidman Cancer Center, University Hospitals Cleveland Medical Center, Cleveland, OH 44106, USA

**Keywords:** antigen presentation, immune checkpoint inhibitors, immunopeptidome, immunoproteasome, proteasome inhibitors, ubiquitin–proteasome system

## Abstract

**Simple Summary:**

Proteasomes are highly complex, macromolecular, protein-degrading machines that execute the controlled elimination of intracellular proteins. Proteasome-dependent protein degradation governs numerous essential cellular processes that regulate cell and circadian cycles, transcription, growth, and development, and execute the efficient removal of abnormal, denatured, and misfolded polypeptides and proteins. Immunoproteasomes represent a highly specialized proteasomal variant that degrades proteins in cells exposed to oxidative stress and proinflammatory stimuli. Immunoproteasomes are significantly elevated in immune cell types and generate oligopeptides that are exhibited on the tumor complexed with MHC class I molecules to facilitate surveillance mechanisms that eradicate cancer cells. Immunoproteasomes represent actionable therapeutic targets that can be pharmacologically manipulated to treat cancer and infectious diseases, as well as proteinopathies characterized by the pathologic accumulation of toxic, proteinaceous aggregates.

**Abstract:**

The majority of T-cell responses involve proteasome-dependent protein degradation and the downstream presentation of oligopeptide products complexed with major histocompatibility complex (MHC) class I (MHC-I) molecules to peptide-restricted CD8^+^ T-cells. However, evasion of host immunity is a cancer hallmark that is achieved by disruption of host antigen processing and presentation machinery (APM). Consequently, mechanisms of immune evasion promote cancer growth and survival as well as de novo and acquired resistance to immunotherapy. A multitude of cell signaling pathways modulate the APM and MHC-I-dependent antigen presentation. Pharmacologics that specifically target and modulate proteasome structure and activity represent a novel emerging strategy to improve the treatment of cancers and other diseases characterized by aberrant protein accumulation. FDA-approved pharmacologics that selectively activate proteasomes and/or immunoproteasomes can be repositioned to overcome the current bottlenecks that hinder drug development to enhance antigen presentation, modulate the immunopeptidome, and enhance the cytotoxic activity of endogenous or engineered T-cells. Strategies to enhance antigen presentation may also improve the antitumor activity of T-cell immunotherapies, checkpoint inhibitors, and cancer vaccines. Proteasomes represent actionable therapeutic targets to treat difficult-to-treat infectious processes and neurodegenerative diseases that are characterized by the unwanted accrual of insoluble, deleterious, and potentially toxic proteins. Taken together, we highlight the breadth and magnitude of the proteasome and the immense potential to amplify and unmask the immunopeptidomic landscape to improve the treatment of a spectrum of human diseases.

## 1. Introduction

Numerous pathways work in coordination to balance the levels of intracellular functional protein and maintain proteostasis in eukaryotic cells [1,2]. Cells preserve key physiological and metabolic processes to remain viable [1,2]. The mechanisms that govern proteostasis are customized to rapidly respond to intracellular perturbations and exogenous stimuli to meet the demands of a complex proteome [3,4,5,6,7]. Under pathologic or stress conditions that promote protein misfolding and denaturation, heat shock proteins and molecular chaperones attempt to refold the protein target. Upon failure of the chaperone machinery, misfolded and unfolded proteins accumulate and trigger endoplasmic reticulum (ER) stress to activate the unfolded protein response (UPR). Activation of the UPR within the ER attempts to limit misfolded protein accumulation and restore proteostasis. Transcriptional activation remodels protein folding and trafficking pathways and modulates protein clearance systems. In the event that protein refolding mechanisms are not successful, protein clearance pathways are activated. The ubiquitin (Ub)-proteasome system (UPS) represents the major proteolytic system in eukaryotic cells and is also activated to remove misfolded proteins. The UPS is a highly adaptable, finely tuned pathway that controls the bulk of cytosolic, nuclear, and specific membrane proteolysis [8,9,10]. While the UPS maintains proteostasis under physiological conditions, it is also deregulated in several pathological states. If the capacity of the UPS is exceeded, aberrant proteins accumulate and form aggregates that may be removed by the autophagy system, most notably macroautophagy. Pharmacologic and genetic modulation of the UPS has emerged as an attractive method to address a growing number of disorders [11,12,13,14,15].

The 26S proteasome is a ~2.5 MDa, ATP-hydrolyzing proteolytic machine that functions as the protein-hydrolyzing catalytic core component of the UPS [16,17,18,19]. 26S proteasomes are assembled in an ATP-dependent reaction and are comprised of two sub-complexes, namely a 20S core particle (CP) and 19S regulatory particles (RPs, PA700) that cap either or both ends of the 20S CP [20,21]. The 20S CP is a barrel-shaped structure that demonstrates multicatalytic proteolytic activity and is comprised of four axially aligned heteroheptameric rings (two outer α- and two inner β-rings) [22,23]. The outer α-rings contain seven structurally similar α-subunits (α1–α7). By forming a pore, the rings function as a tightly regulated gate for the entrance of substrates, and for the removal of degradation products. The inner β-rings contain catalytically active peptide-hydrolyzing subunits. Mechanisms of gate opening and proteasome activity are regulated by docking of regulator proteins, e.g., 19S RPs, PA28, PA200, ECM29 and PI31 [24]. In addition, a number of endogenous proteins have been identified that are loosely associated with the proteasome and regulate proteolytic activity [24,25,26]. For example, the HR23B/Rad23B targets the UPS where it shuttles ubiquitinated cargo to proteasomes that are destined for subsequent degradation.

Although intracellular protein degradation pathways are tightly regulated, proteasomes are highly abundant in most eukaryotic cells. Crystallography of purified 20S proteasomes has shown that the middle of the *α*-ring is nearly entirely occluded to hinder substrates from entering the inner core surrounded by proteolytically active *β*-subunits [25,26]. In addition, the amino region of the *α*-subunits blocks substrate entry to the β-ring catalytic sites. Hence, while highly abundant in eukaryotic cells, 20S proteasomes are maintained in a restrained state. The association of polyubiquitinated substrates to 19S RPs of mammalian and yeast 26S proteasomes enhanced 20S proteasome peptidase activities ~2-fold in a process that required ATP hydrolysis [27]. However, 26S proteasomes from the yeast α3ΔN open-gate mutant and the *rpt2YA* and *rpt5YA* mutants with impaired gating were still activated (1.3–1.8-fold) by polyUb-protein binding. Thus, the binding of polyubiquitinated substrates to the 19S RP stabilizes the gate opening of the 20S proteasome and induces conformational changes of the 20S proteasome that facilitate the channeling of substrates and their access to active sites. Polyubiquitinated substrates allosterically stimulate their own degradation.

The Ub-binding receptors, e.g., Rpn10 and Rpn13, have the ability to bind protein substrates that bear ubiquitin chains [28]. Several receptors and accessory proteins are attached to proteasomes through labile interactions and participate in modulating the functional activity of proteasomes [28]. Specific Ub-binding receptors shuttle ubiquitinated cargo destined for proteasomal degradation, but are not inherent proteasomal constituents [29,30,31,32,33,34,35,36,37,38,39,40,41,42]. Many shuttle proteins, e.g., HR23A/B, possess an N-terminal Ub-like (UbL) region and C-terminal Ub-associated (UBA) domains, which empower these proteins to recognize and transport protein substrates that bear poly-Ub chains.

A number of endogenous proteasome activators have been described [43]. The 11S/PA28/REG family of proteasome activators extends across all multicellular cells. These activators are ATP-independent, heptameric complexes that do not contain unfolding or forced translocation activity [43]. There are three known homologs of PA28-α, β, and γ. PA28α and PA28β form an asymmetric heteroheptameric complex known as PA28αβ. PA28αβ expression is regulated by IFN-γ and functions in MHC-I antigen presentation. PA28γ is a homoheptameric complex that does not form a complex with PA28α or PA28β, and is implicated in a variety of disease states. Proteasomal ATPases, e.g., 19S RPs, utilize ATP to recognize, unfold, and translocate substrates into the degradation chamber. ATP-independent proteasome activators include PA200/Blm10, PI31, and 11S, each with its own unique regulatory mechanisms [43].

## 2. Immunoproteasomes

Proteasomes display extensive structural heterogeneity and proteasomal subtypes can be classified based upon their constituent catalytic subunit composition. Two main forms of the proteasome exist, namely the constitutive proteasome and the immunoproteasome [44,45]. Constitutive proteasomes contain β1, β2, and β5 catalytic subunits which are expressed in nearly all human cells. Immunoproteasomes contain the β1i, β2i, and β5i catalytic subunits and under proinflammatory conditions, cells switch to express immunoproteasome subunits over constitutive subunits [44,45,46,47]. The immunoproteasome subunits substitute for constitutive subunits during proteasome assembly. Immunoproteasomes are constitutively expressed in dendritic cells and antigen-presenting cells [44,45,46]. Catalytic β1, β2, and β5 subunits are replaced by the subunits β1i (low molecular mass protein-2, LMP2), β2i (multicatalytic endopeptidase complex-like 1, MECL-1) and β5i (low-molecular mass protein-7, LMP7), respectively [46,47,48]. Consequently, proteasomes and immunoproteasomes cleave protein substrates differently and display distinct sensitivities to small molecule inhibitors [49,50]. Immunoproteasomes show a higher cleavage preference after hydrophobic and basic residues. Upon exposure to interferon gamma (IFN-γ) and other cytokines, two β-type subunits are replaced by proteins encoded in the MHC. Interestingly, a third subunit, MECL-1, is also induced by IFN-γ but is encoded outside the MHC. The essential co-incorporation of MECL-1 and LMP2 is a crucial factor to consider when interpreting results obtained from cell lines and mice lacking LMP2, and in studies that address the role of MECL-1 in MHC-I presentation [51,52]. Immunoproteasomes have been optimized to produce antigens for presentation by MHC-I molecules and most CD8^+^ T-cell epitopes result from immunoproteasome cleavage of protein substrates. Peptides generated by the proteasome are generally 8–10 amino acids in length and mirror the linear sequence of the parental protein. Given that proteasomes and immunoproteasomes determine the repertoire of CD8^+^ T-cell epitopes, researchers have developed different approaches to predict proteasomal cleavage sites [53,54,55,56]. Since the C-terminus of peptides presented by MHC-I molecules corresponds to the P1 residue of the cleavage site, models have been generated to predict proteasome cleavage sites [53,54,55,56].

Immunoproteasomes degrade viral, bacterial, and tumor proteins to generate small peptide products, some of which are presented as antigens complexed with MHC-I molecules. The presentation of viral or tumor-specific antigens triggers CD8^+^ T-cells to initiate robust immune responses as a form of cell-mediated immunity [57,58,59]. Immunoproteasomes are relevant not only to cancer but also to infectious diseases. Immunoproteasomes also serve a key role in protecting cells when exposed to oxidative stress induced by IFNs. IFNs stimulate the generation of reactive oxygen species (ROS), leading to protein oxidation and damage. Concurrently, the ubiquitylation of protein substrates is elevated in response to IFNs [60,61]. Prior studies in murine systems have demonstrated that immunoproteasomes also prevent the aggregation of damaged proteins. Functionally active immunoproteasomes in non-immune cell types and tissues under healthy and pathological states challenge the notion that immunoproteasomes solely function for MHC-I antigen presentation [62].

## 3. Targeting MHC Class I Antigen Presentation Machinery to Boost T-Cell Responses

Proteasomes function not only as the catalytic core of the UPS as the primary protein degradation system in eukaryotic cells but also are largely responsible for the processing of antigens for presentation by the MHC class I pathway [63]. Antigenic peptides are generated from the proteasomal degradation of endogenously synthesized proteins or exogenous proteins acquired by host cells [64]. Endogenous antigen presentation occurs constitutively in nearly all cell types and is enhanced by IFNs and other cytokines. Defective Ribosomal Products (DRiPs) are a subset of rapidly degraded polypeptides that do not attain a stable conformation but rather are degraded during or shortly after translation [64]. DRiPs have been defined as prematurely terminated polypeptides and misfolded polypeptides produced from the translation of bona fide mRNAs in the proper reading frame [65]. Importantly, DRiPs provide a direct linkage between translation and peptide generation to enable the rapid recognition of virus-infected cells and immunosurveillance of acute changes in cellular gene expression. Rapidly degraded translation products, DRiPs, serve as a primary source of peptide precursors to optimize immunosurveillance of pathogens and tumors.

The development of immunotherapeutics focused on T-cell cytotoxic activity is a powerful tool to fight cancer. The premise for such a strategy is that the phenotypic alteration of tumor cells and repertoire of tumor-associated antigens (TAAgs) can be targeted by effectors of the host immune system. However, evasion of immunosurveillance is a hallmark of cancer that thwarts the efficacy of immunotherapeutics and reduces patient benefit [66]. Newly emerging results indicate that the successful development of antitumor immune responses is linked to T-cells that target cancer cell-specific neoepitopes. These neoepitopes are a category of peptides bound to the MHC-I and result from tumor-unique mutations. However, identifying actionable neoantigens (NeoAgs) exclusively presented on cancer cells remains a challenge [67,68,69]. Immunoproteasome upregulation is linked with improved response to treatment with immune checkpoint inhibitors (ICIs) [70,71]. These observations suggest the need for highly sophisticated proteolytic machinery with the functional plasticity to generate finely tailored products. Novel small molecules and pharmacologics that activate proteasomes represent a tractable approach to increase antigen presentation, overcome immune escape, and enhance T-cell cytotoxic activity (Figure 1).

Immune checkpoints are constituent components of a healthy human immune system and function to prevent an immune response from being so strong that it destroys healthy (self) cells. Immune checkpoints are engaged when proteins on the T-cell surface, e.g., PD-1, CTLA-4, recognize and bind to partner proteins, e.g., PD-L1/PD-L2, on tumor cells. Therapy that releases the natural brake of the immune system and functionally blocks the immune checkpoint, i.e., immune checkpoint blockade (ICB), has demonstrated long-term survival benefits as monotherapy. Nonetheless, only a relatively small fraction of patients benefit from it due to reduced MHC-I expression, limited NeoAg level, *HLA* heterozygosity loss, and prolonged exposure to IFN-γ [72]. Gu et al. employed genome-wide CRISPR screening to identify drugs that upregulated MHC-I without inducing PD-L1 and identified a Second Mitochondria-derived Activator of Caspase (SMAC) mimetic [73]. The SMAC mimetic birinapant enhances the expression of MHC-I, increases the susceptibility of cancer cells to T-cell-mediated destruction, and contributes to the effectiveness of ICB. The cyclin-dependent kinase (CDK) 4/6 inhibitors lead to cell-cycle arrest but also trigger T cell-mediated immunity. The effects of the CDK4/6 inhibitors abemaciclib and palbociclib on antigen presentation in breast cancer cells were evaluated by identification of HLA ligands [73]. The CDK4/6 inhibitor treatment upregulated HLA and revealed hundreds of HLA ligands. MAP2K1 (MEK), EGFR, and RET were validated as negative regulators of MHC-I expression and APM in many cancers [74]. BRAF inhibitors also enhanced melanoma antigen expression and increased T-cell activity [75].

Tumors have developed mechanisms to render themselves invisible to immunosurveillance as an immunologically “cold” tumor phenotype. HLA-I deficient cells are characterized by T-cell epitopes that are associated with impaired peptide processing (TEIPP), derived from non-mutated polypeptides that are not typically antigenic [76]. Primary and acquired resistance to ICB therapy was associated with alterations in genes relevant to antigen presentation by MHC-class I/β-2-microglobulin (MHC-I/β2m) complexes to CD8 T lymphocytes [77]. Cancer cells evade CTL recognition and killing through alterations in TAP that play a major role by inducing a sharp decrease in surface expression of MHC-I/β2m-peptide complexes, enabling malignant cells to become “invisible” to CD8 T cells. Downregulation of MHC-I molecules has been found to range from 16 to 50% among primary lesions from various types of human carcinomas [78]. Moreover, between 39 and 88% of human tumors were reported to be MHC-I deficient, including 73% of lung cancers with a total loss of class I molecules in 38% and loss of A locus and A2 allele in 8.3 and 27% of the analyzed cases, respectively [79]. Downregulation of TAP1 and/or TAP2 in lung cancer cells, resulting in resistance to TCR-dependent lysis [80]. TAP deficiencies have been observed in a wide variety of human cancers, including cervical carcinoma [81], head and neck carcinoma [82], melanoma, gastric cancer [83,84], and lung cancer, with up to 70% of NSCLC expressing low levels of TAP1 and/or TAP2. In addition to the increase in HLA expression, the ALK and RET inhibitors could also alter the cell’s protein repertoire independent of the effects on antigen presentation pathway throughput, thus providing potential new antigens. The inhibited ALK and RET kinases are upstream of multiple signaling pathways that control the expression of multiple target genes. This could lead to the appearance of the new peptides found in the drug-treated groups, which could potentially include tumor-associated antigens.

The benefits of T-cell immunotherapies are often hindered by reduced presentation of tumor-specific antigens abetted by the downregulation of human leukocyte antigen (HLA). Drugs inhibiting ALK and RET produced dose-related increases in cell-surface HLA in tumor cells bearing these mutated kinases in vitro and in vivo, as well as elevated transcript and protein expression of HLA and other antigen-processing machinery [85]. Subsequent analysis of HLA-presented peptides after ALK and RET inhibitor treatment identified large changes in the immunopeptidome with the appearance of hundreds of new antigens, including T-cell epitopes associated with impaired peptide processing (TEIPP) peptides. T-cell epitopes associated with impaired peptide processing, e.g., TEIPP peptides, were highly elevated [86]. An increase in the expression of HLA can make those cells preferable targets for T-cell–based immunotherapies. Inhibitors of HDAC, DNA methylase, ALK, RET, and the MAPK pathway have been shown to influence HLA expression levels. MAPK inhibitors increase HLA levels in a STAT1-dependent manner. Inflammatory cytokines, e.g., IL-6, IL-12, IFN-γ, and TNF-α, have been shown to also influence HLA levels.

Nascent HLA molecules reside in the ER until they associate with β-2-microglobulin, after which TAP1 and TAP2 transport the proteasome-cleaved peptides into the ER and antigenic peptides are loaded onto the complex. Conceivably, this could lead to increased transcription or translation, increased stabilization by peptide loading and β-2-microglobulin association, or reduced degradation. Drugs also caused variable increases in transcript levels of HLA and other proteins involved in antigen processing machinery, though in general there was an increase in either HLA and/or transporter-associated antigen processing (TAP) TAP1, TAP2, or β-2-microglobulin. Increased TAP1, TAP2, and β-2-microglobulin levels support enhanced peptide loading in the ER and more stabilization of the cell-surface HLA.

The lysine acetyltransferase p300/CREB binding protein (CBP) has been shown to modulate MHC-I antigen processing and presentation [87]. Treatment with the DNA-damaging platinoid oxaliplatin and the topoisomerase inhibitor mitoxantrone upregulated MHC-I antigens in a process dependent on the nuclear factor-kappa B (NF-κB). Ablation of NF-κB and p300 precluded the effects of DNA-damaging agents on antigen presentation, abolished the effects on tumor cells, and reinvigorated CD8^+^ cytotoxic T-cells (CTLs). Treatment with oxaliplatin and mitoxantrone may overcome resistance to PD-(L)1 inhibitors in tumors that have downregulated, but not completely lost, components of the APM.

Antigenic peptides generated by proteasomes in the cytosol are delivered to the ER by the TAP1–TAP2 peptide complex [87,88]. The MHC-II-encoded dimeric transporter consists of the TAP1 and TAP2 proteins, and plays a crucial role in the loading of viral peptides onto MHC-I molecules. It was shown that the proteasome genes *PSMB9* and *PSMB8* are situated near the *TAP* genes in the MHC class II region. Tumors may display particularly low levels of the TAP1–TAP2 transporters and tapasin [89]. Heterodimers of the MHC-I glycoprotein and β_2_-microglobulin (β_2_m) also bind short peptides within the ER. Before peptide binding, these molecules form a multisubunit loading complex that contains the TAP subunits, the transmembrane glycoprotein tapasin, calreticulin, and the thiol oxidoreductase ERp57 [90].

Schmidt et al. focused on the functional role of the ER-situated aminopeptidases ERAP1 and ERAP2 in TCR-mediated tumor cell recognition [91]. The authors studied three human HLA-A* 02:01-presented T-cell epitopes and found that ERAP2 alone, when expressed in ERAP-deficient cells, generated a potent response against the tyrosinase_368–376_ epitope. TAP-dependent, N-terminally extended epitope precursor peptides generated in vitro were differentially processed by ERAP1 and ERAP2 and may therefore serve as a source for the tyrosinase_368–376_ epitope. ERAP2 also influenced the recognition of the gp100_209–217_ tumor epitope and enhanced T-cell recognition of the MART-1_26/27–35_ epitope in the absence of *ERAP1*. The results highlight the functional role of ERAP2 in tumor epitope presentation and TCR recognition.

## 4. Harnessing Proteasomes to Increase Neoantigens and Tumor-Associated Antigens

An improved understanding of how the repertoire of tumor-specific antigens—the immunopeptidome—can be modulated is intended to improve current and future anticancer strategies [92,93,94]. Also, it remains unclear how the tumor immunopeptidome changes during disease progression. Moreover, understanding how current standard-of-care therapies alter NeoAg and TAAg presentation is highly valuable. NeoAgs are newly formed antigens generated by mutation specifically within tumor cells, as well as by post-translational modification, RNA splicing, and integration of viral open reading frames (ORFs) [92,95,96]. Importantly, NeoAgs are not subject to central and peripheral tolerance. The prediction and identification of tumor-specific NeoAgs can achieved through the application of next-generation sequencing (NGS) and bioinformatic tools. NeoAgs are promising targets to personalize cancer therapy and may predict survival prognosis and response to ICB. Immunotherapeutics can also be developed against public NeoAgs derived from recurrent mutations in cancer driver genes. A current problem is the prediction and identification of MHC-I peptides that modulate T-cell antitumor responses.

Rana et al. identified FDA-approved bioactive molecules that enhanced proteasome activity and demonstrated that histone deacetylase 6 (HDAC6) inhibitors increased proteasomal hydrolysis of small fluorogenic proteasome substrates, increased pan-MHC-I antigen presentation, and enhanced the anti-myeloma cytotoxic activity of autologous patient-derived T-cells [97]. High-throughput screening of compound libraries has identified modulators of proteasome and immunoproteasome activity (Figure 2). Treatment of multiple myeloma (MM) cells with a panel of FDA-approved and investigational HDAC6 inhibitors dramatically increased MHC-I antigen presentation and significantly enhanced the antimyeloma activity of healthy and patient-derived autologous T-cells. In contrast, treatment of MM cells with the FDA-approved proteasome inhibitor bortezomib dramatically reduced the presentation of MHC-I antigens. While proteasome inhibitors are the backbone of standard-of-care antimyeloma regimens, the effect on endogenous T-cell activity is not completely defined and may actually reduce immune responses [98,99,100,101].

Nearly 90% of newly diagnosed cancers are solid tumors, but very few drugs generate durable sustained responses. Bispecific T-cell engagers (BiTEs) are a well-studied form of T-cell-engager therapy that binds to TAAgs on tumor cells and CD3 on T-cells. BiTEs redirect polyclonal T-cells to form immune synapses leading to tumor cells independent of TCR-mediated TAAg recognition. BiTEs have shown efficacy in the treatment of relapsed and/or refractory B-cell-precursor acute lymphoblastic leukemia (ALL) and relapsed and/or refractory MM (RRMM), with therapeutic effects similar to those reported with chimeric antigen receptor (CAR)-T-cell therapies.

## 5. Targeting Proteasomes to Treat Infectious Diseases

Worldwide, cancer and infectious disease are major causes of morbidity and mortality, and while the causative agents and molecular mechanisms of disease may differ, they share many pathophysiologic similarities [19,102,103,104]. Proteasomes are conserved throughout evolution and contribute to the cell biology and viability of parasites, and unicellular and multicellular organisms [105,106,107]. These observations highlight the importance of proteasomes and regulated protein degradation. Cancer cells and the organisms responsible for infectious disease are comprised of cell populations that undergo selective pressure upon drug challenge and develop drug resistance which impacts patient response to treatment and eventual outcome. Similar to the active sites that reside within catalytic subunits of proteasomes within human cells, proteasomes in pathogenic organisms, e.g., *Mycobacterium tuberculosis* and *Plasmodium falciparum*, also possess a nucleophilic Thr residue within the active site [104,105,106,107]. Consequently, certain pathogenic organisms are sensitive proteasome inhibitors, including bortezomib, carfilzomib, and ixazomib which have been FDA-approved to target human proteasomes for the treatment of cancer. Cancer cells and infectious organisms also express certain common proteins that can be recognized by effectors of host immunity and both disease types provoke inflammation and elicit T-cell-driven immunologic responses.

Analogs of the selective proteasome inhibitors bortezomib and carfilzomib have been identified to target proteasomes from *Mycobacterium tuberculosis* (Mtb) and *Plasmodium falciparum* pathogens [108,109]. Cell-based chemical screening methods have identified potent inhibitors of proteasomes found in *Leishmania* and *Trypanosoma* species [103,107,110]. Carmaphycin B is a naturally derived molecule and a potent inhibitor against both the asexual and sexual blood stages of malaria infection [110]. Using a combination of in silico molecular docking and in vitro directed evolution in a well-characterized drug-sensitive yeast model, compounds targeted the β5 subunit of the proteasome. These studies were validated using in vitro inhibition assays with proteasomes isolated from *Plasmodium falciparum*. Since carmaphycin B is toxic to mammalian cells, the authors synthesized a series of chemical analogs that reduce host cell toxicity while maintaining blood-stage and gametocytocidal antimalarial activity and proteasome inhibition. Lactacystin is a natural product synthesized by *Streptomyces* and has to be used extensively in cell-based assays with kinetoplastid parasites. A high-throughput phenotypic study performed at the Genomics Institute of the Novartis Research Foundation identified a compound termed GNF5343 as a hit in proliferation assays of *L. donovani* and *T. brucei* with half maximal effective concentration (EC_50_) values of 7.3 and 0.15 μm, respectively [107]. GNF5343 is an azabenzoxazole that was identified as a potent inhibitor of *Leishmania donovani* and *Trypanosoma brucei* proliferation from a screen of 3 million compounds. GNF6702 was optimized from GNF5343 to have improved bioavailability and potency. GNF6702 is a broad-spectrum antiprotozoal drug that acts as a non-competitive proteasome inhibitor, effective against infection with any of the three protozoal parasites in mice but displaying minimal toxicity in mammalian cells [107]. GNF6702 displays unprecedented in vivo efficacy, clearing parasites from mice in all three models of infection. GNF6702 inhibits the kinetoplastid proteasome through a non-competitive mechanism, does not inhibit the mammalian proteasome or growth of mammalian cells, and is well tolerated in mice.

HLA allelic diversity is required to expand the repertoire of MHC-I peptides that are presented to the T-cell population. Consequently, HLA molecules are highly polymorphic and contribute to host responses to foreign pathogens. Prior work has linked the expression of specific HLA alleles and haplotypes with the susceptibility of humans to infection of SARS-CoV-2, the etiological agent responsible for the COVID-19 pandemic [111]. Conversely, protective HLA variants have also been identified for both mild and severe forms of SARS-CoV-2 infection. Given the pivotal function of HLA-mediated immunity in COVID-19 infection, an increasing number of reports have linked HLA variants to diverse COVID-19 consequences and identified HLA genotypes that may influence distinct immunologic reactions. Upon entry of SARS-CoV-2 into the body, the virus triggers immune cells to produce significant amounts of IFN-γ. In turn, the expression of the proteasome activator PA28γ is elevated [112]. Consequently, proteasome activity is increased and promotes degradation of the coronavirus N protein, leading to reduced viral production and limited proliferation. Increased levels of individual proteasome subunits in patients with COVID-19 suggest that increased proteasome activity may represent an approach to drive T-cell responses.

Antigenic drift denotes the amino acid substitutions in viral proteins that evolve and accumulate upon selection by host adaptive immunity as the virus disseminates throughout a population [113]. The evolutionary accumulation of amino acid changes can limit the duration of immunity conferred by infection and vaccination. The spike protein of SARS-CoV-2 has been reported to rapidly undergo antigenic drift. Following the initial public reporting of SARS-CoV-2 in 2019, studies have identified multiple immunodominant CD8^+^ T cell epitopes of SARS-CoV proteins. Several studies have shown how mutations within the epitopes of SARS-CoV-2 proteins lead to viral escape from T-cells. Studies of the human immunodeficiency virus (HIV-1) showed that mutations in HIV proteins disrupted proteasomal cleavage and degradation However, a precise understanding of the effect of SARS-CoV-2 mutations on viral epitope antigen processing and CD8^+^ T cell activation is lacking [114].

## 6. Targeting Proteasomes to Treat Neurodegenerative Diseases

Proteasomes play a vital role in the functioning and survival of neuronal cells. Deregulation of proteasome activity has been reported during neurodegenerative processes, i.e., the deposition of protein aggregates within the cell [115,116]. Immunoproteasome *LMP2* and *LMP7* subunits have been systematically detected in brain biopsies isolated post-mortem from subjects diagnosed with neurodegenerative conditions, e.g., Alzheimer’s disease (AD) and Huntington disease (HD) as well as in the cerebral cortex and hippocampus of a rodent model [117]. The characteristic buildup of Ub-conjugated proteins in AD and HD indicates that extensively aggregated proteins disrupt proteasomal activity [118]. A common feature of HD is the buildup of Lys48-, Lys11-, and Lys63-linked Ub chains in HD mouse models, as well as the brains of HD patients. The E3 Ub ligase tumor necrosis factor receptor-associated factor 6 (TRAF6) was also overexpressed in postmortem brain tissue of HD patients. Parkinson’s disease (PD) is associated with mutations in several proteins that have been linked with proteasomes, including alpha-synuclein (α-SNCA), protein deglycase DJ-1 (PARK7), UCHL1, PTEN-induced kinase 1 (PINK1), and PD protein 2 (PARK2, parkin) [119,120].

The rare neurological disease amyotrophic lateral sclerosis (ALS), Lou Gehrig’s Disease, affects motor neurons and is characterized by the progressive degeneration of the brain and spinal cord [119]. Several mutations have been linked to the familial form of ALS, including alteration of superoxide dismutase 1 (SOD1), ubiquilin 2 (UBQLN2), and RNA-binding protein fused in sarcoma (FUS) [120]. As early as the 1960s, studies had demonstrated protein inclusions were identified in the anterior horn cells of patients with the sporadic and familial forms of ALS [121]. Therapeutic interventions in neurodegenerative diseases are needed since the pathophysiology of such diseases is not completely understood. Importantly, the UPS is disrupted in ALS patient samples as well as experimental models of ALS. Abnormal protein accumulations are a hallmark of ALS and these accumulations contain TDP-43 [122], neurofilament [123], FUS [124], or SOD1 [125]. TDP-43 is present in up to 98% of the accumulations in sporadic and familial cases [126]. Strategies to activate proteasome-mediated clearance of protein inclusions characteristic of ALS represent a novel therapeutic option.

Recently, three different proteins from AD, PD, and HD that share a common three-dimensional structure were shown to potentially impair proteasomes [127,128,129]. The shared conformation allowed these proteins to bind the proteasome with low nanomolar affinity, leading to impaired Ub-dependent and Ub-independent activity. Studies demonstrated that these oligomeric proteins also allosterically impaired substrate entry by blocking the gate of the 20S CP and preventing the 19S RP from injecting substrates into the degradation chamber. These results provided a novel molecular model for oligomer-driven impairment of proteasome function. Lee et al. reported that linear peptide epoxyketones targeting the immunoproteasome may represent a new class of AD drugs to ameliorate cognitive deficits, independently of Aβ or tau accumulation [130]. While displaying promising efficacy, the prospect of these linear peptide epoxyketones for clinical use in AD appears limited due to poor brain accessibility, in vivo metabolic instability, and short circulation time. ABCB1-mediated drug efflux and hydrolysis by peptidases and epoxide hydrolases may contribute to the lack of substrate accessibility. Peptide epoxyketones, short peptides with a C-terminal α′, β′-epoxyketone warhead, offer pharmacologic advantages conferred by their proven target specificity for the proteasome.

The 20S proteasome is the main protease for the degradation of oxidatively damaged and intrinsically disordered proteins. When the accumulation of disordered or oxidatively damaged proteins exceeds proper clearance in neurons, imbalanced pathway signaling or aggregation occurs, both of which have been implicated in the pathogenesis of several neurological disorders. Screening of the NIH Clinical Collection and Prestwick libraries identified the neuroleptic agent chlorpromazine as a lead agent capable of enhancing 20S proteasome activity. Separately, Jones et al. have shown that chlorpromazine increases 20S proteasome ChT-L activity and the degradation of α-synuclein and Tau [131]. Similarly, Trader and colleagues revealed that MK-886 and AM-404 can act as 20S activators to improve α-synuclein degradation in culture [132]. A screen of the NIH Clinical Collection (NCC) revealed that oleuropein and betulinic acid increased hydrolysis of a proteasome fluorogenic substrate [131]. In addition, two small molecules from the NCC, MK-866, and AM-404, stimulated the proteasome-mediated turnover of a misfolded protein in living cells by three- to four-fold. Proteasomal catalytic activity decreases with age to consequently increase the risk of diseases characterized by protein aggregation.

## 7. Proteasomes and Aging

Aging can be described as the sum of all the events that change the functions of a living cell or organism, prevent it from maintaining physiological balance, and eventually culminate in the total cessation of all biological processes, i.e., death. Aging is a normal, gradual, continual process of natural change [133,134,135]. The capacity of cells to maintain proteostasis and a healthy proteome gradually declines during the aging process [134,136,137,138,139,140]. However, the functionality of the mechanisms that maintain proteostasis also gradually declines with aging and leads to a number of age-related diseases. The observed decline in proteostasis with aging also involves a decrease in the stability of properly folded proteins and a significant reduction in the efficiency of protein clearance. During aging, the global activity of the UPS declines in multicellular organisms through mechanisms that remain poorly understood. More pointedly, aging is paralleled by a decline in proteasome activity and, consequently, has also been associated with the accumulation of damaged proteins [139,140,141,142,143,144,145]. Proteasomal dysfunction is manifest through a reduction in the level of individual subunit levels, as well as the modification or substitution of proteasome subunits, proteasome disassembly, and proteasome inactivation [141,142,143]. As a consequence of aging, environmental stress, or lack of functional requirements, proteins undergo degradation to limit the threat raised by their maintenance. The progressive exposure of stressors during aging induces the accumulation of damaged and unfolded proteins which impairs the canonical protein degradation pathways, e.g., the UPS. Oxidized and structurally misfolded proteins accumulate with aging and eventually impair essential cellular functions.

An important goal of aging research is to define the cellular, molecular, and genetic changes in the effectors of protein degradation that occur and to then identify actionable therapeutic targets [144]. Importantly, a distinction between the causes and consequences of aging on protein degradation and proteasomes is essential for a better understanding of aging. In addition, the recognition of short-term consequences or responses needs to be distinguished from long-term pathway-specific adaptations. Commonly recognized fundamental mechanisms of the aging process are critically examined for the possibility of their adaptation-driven emergence from processes such as cell competition and the wound-like features of the aging body.

A better understanding of the biology of aging and its impact on the prevention, progression, and prognosis of disease and disability remains crucial. An intriguing, diverse array of strategies to overcome the age-associated decline in proteasome activity have been proposed to genetically or pharmacologically increase proteasome-associated activity. Another approach is to decrease the formation of precursor proteins in an attempt to slow the accumulation of protein aggregates and pathologic plaques and to limit disease initiation or arrest disease progression. In yeast, defects in the UPS have been identified during aging that also result in the generation of protein inclusions [145]. By increasing disaggregase activity, the number of age-related protein inclusions is reduced [146]. Disaggregation of abnormal protein aggregates may enhance proteasome activity without increasing proteasome levels.

## 8. Proteasomes and Autophagy

Proteostasis refers to the regulation of the cellular concentration, folding, interactions, and localization of each of the proteins that comprise the proteome. Lysosomes are an important site for the degradation of misfolded proteins, which are trafficked to this organelle by the pathways of macroautophagy, chaperone-mediated autophagy, and endocytosis. With the discovery of the lysosome by Christian de Duve, it was assumed that cellular proteins are degraded within this organelle [147]. However, it was later demonstrated that protein degradation was a selective process. In addition, several independent lines of evidence suggested that intracellular proteolysis was largely non-lysosomal, through mechanisms that remained obscure [148]. The heterogeneous stability and turnover rates of individual proteins, as well as the effect of nutrients and hormones on their degradation and the dependence of intracellular proteolysis on metabolic energy, strongly suggested other mechanisms of protein degradation.

Eukaryotic cells have two major intracellular protein degradation pathways, namely the UPS and autophagy [149,150,151]. The UPS and autophagy-lysosome pathway (ALP) have long been thought of as separate, parallel catabolic systems. However, emerging evidence has shown that the UPS and autophagy serve in a complementary manner to each other to mediate the degradation of polyubiquitinated proteins and to resolve ER stress. Autophagy is a major intracellular degradation system that derives its degradative abilities from the lysosome [149,150,151]. The most well-studied form of autophagy is macroautophagy, which delivers cytoplasmic material to lysosomes through a double-membraned autophagosome. Autophagy controls important physiological functions where cellular components need to be degraded and recycled. Autophagy can rapidly provide fuel for energy and building blocks for renewal of cellular components, and is therefore essential for the cellular response to starvation and other types of stress. Extensive crosstalk has been identified between the UPS and the ALP in human cells [148,149,150,151]. When the UPS was impaired, the ALP was induced to compensate for the limited activity of the UPP. When the ALP was compromised, the UPP was also upregulated. Cooperation between the two pathways plays an essential role in eliminating misfolded and damaged proteins.

Recent studies demonstrate that the activities of the UPS and ALP are connected and that there is substantial crosstalk between the two protein clearance pathways [152,153]. Ubiquitination is utilized as a signal for degradation by both pathways. Inhibition of proteasomal activities has been previously shown to induce autophagy, indicating a coordinated and complementary relationship between these two systems [152]. Lysosomotropic agents, e.g., chloroquine, impair autophagy and result in weak proteasome inhibition or proteasome overload. Also, the limited accumulation of ubiquitinated proteins was observed. Another study demonstrated that autophagy inhibition increased levels of proteasome substrates, primarily due to p62/SQSTM1 accumulation after autophagy inhibition [154]. Excess p62 inhibits the clearance of ubiquitinated proteins destined for proteasomal degradation by delaying their delivery to the proteasome. The results also showed that the inhibition of autophagy, previously thought to only affect long-lived proteins, also compromised the UPS. In contrast, proteasomes were reported to be activated in response to pharmacological inhibition of autophagy as well as disruption of autophagy-related genes by RNA interference under nutrient-deficient conditions in cultured human colon cancer cells [151]. The induction was evidenced by the increased proteasomal activities and upregulation of proteasomal subunits, including the proteasome β5 subunit, PSMB5. Co-inhibition of the proteasome and autophagy also synergistically increased the accumulation of polyubiquitinated proteins.

It is reasonable to assume that the two arms of protein degradation may compensate for each other upon the inhibition or deregulation of one arm. Therefore, there is a rationale to simultaneously target both protein clearance pathways pharmacologically for cancer treatment [155,156]. Plasma cells are professional antibody factories that secrete vast amounts of immunoglobulins (Igs) and myeloma cells are exquisitely sensitive to disruption of protein homeostasis [157]. Proteasome inhibitors target the unfolded protein response by inhibiting the degradation of ubiquitinated proteins (and paraproteins), leading to protein accumulation within the ER that culminates in cell death. The combined effects of hydroxychloroquine (HCQ) and the reversible proteasome inhibitor bortezomib or the irreversible inhibitor carfilzomib were tested on MM cell lines and primary cells [158]. As expected, HCQ potentiated carfilzomib-induced myeloma cell death, but surprisingly, HCQ had little or no effect on bortezomib activity. HCQ also potentiated the effect of the irreversible proteasome inhibitor oprozomib, but not the effect of the reversible inhibitor ixazomib. Thus, the sensitivity towards the irreversible proteasome inhibitors carfilzomib or oprozomib is to a larger extent affected by the autophagic system than the reversible inhibitors bortezomib and ixazomib. High-throughput screens to detect pharmacologics that modulated autophagy to enhance the anti-myeloma effect of bortezomib revealed metformin, a widely used antidiabetic agent with proven efficacy and limited adverse effects [159]. Metformin co-treatment with bortezomib suppressed induction of the critical UPR effector glucose-regulated protein 78 (GRP78) to impair autophagosome formation and enhance apoptosis. Gene expression profiling of newly diagnosed myeloma patient tumors further correlated the hyperexpression of GRP78-encoding *HSPA5* with reduced clinical response to bortezomib. The effect of bortezomib was enhanced with metformin co-treatment using myeloma patient tumor cells and the chemoresistant, stem cell-like side population that may contribute to disease recurrence.

## 9. Modifiers of Proteasome Activators to Boost Immunotherapeutic Responses

Modulating the level of individual target proteins that are degraded by the UPS has recently expanded the scope of pharmacological inventions in cancer and immuno-oncology [160,161]. Stimulator of interferon genes (STING) is an auspicious target for immunotherapy and studies have demonstrated the importance of STING as well as the utility of its agonists in immunotherapy outcomes. A number of STING agonists have demonstrated promising biological activity and showed excellent synergistic anti-tumor effects in combination with other cancer therapies in preclinical studies and some clinical trials. Post-translational modifications, including phosphorylation, ubiquitination, palmitoylation, and SUMOylation, have been reported to play an essential role in regulating STING function [162]. STING is an ER-associated transmembrane protein that turns on and quickly turns off downstream signaling as it translocates from the ER to vesicles [163]. Blockade of trafficking-mediated STING degradation using the macrolide antibiotic and autophagy inhibitor bafilomycin A1 specifically enhanced cGAMP-mediated immune response and anti-tumor effect in mice. The proteasome inhibitor bortezomib had a slight effect on STING levels. Hence, trafficking and sorting of substrates, e.g., STING to acidified endolysosomes, can be targeted to enhance antitumor responses.

The E3 ligases tripartite motif containing 56 (TRIM56), TRIM32, TRIM10, and autocrine motility factor receptor (AMFR) have been proposed to catalyze K63-, K27-, or K29-linked polyubiquitination to boost STING signaling [164]. The small molecule SB24011 inhibits STING-TRIM29 E3 ligase interaction and therefore prevents TRIM29-induced degradation of STING. SB24011 was shown to enhance STING immunity by upregulating STING protein levels, which robustly potentiated the immunotherapy efficacy of STING agonist and anti-PD-1 antibody via systemic anticancer immunity. Proteasome inhibitors such as bortezomib and carfilzomib that globally block the UPS may not be appropriate for the selected targeting of individual proteins like STING. However, inhibitors of specific E3 Ub ligase and proteolysis targeting chimeras (PROTACs) technology may be appropriate [165].

## 10. Conclusions

A variety of distinct strategies have been developed to increase proteasome activity with the intent to treat a spectrum of human maladies. Proteasome activation offers the potential to modulate global or selective protein degradation to treat cancers, as well as infectious and neurodegenerative diseases, and to ameliorate the deleterious effects of aging. Thus, future clinical applications of proteasome activators may be context-dependent and require disease-specific therapies. The potential for adverse events may be dictated by the precise mechanism of proteasome activation rather than global effects on the UPS. Recent results highlight the breadth and magnitude of the proteasome in governing fundamental cellular processes and the immense potential of therapeutics that exploit proteasomes to treat disease.

Immunotherapeutics have been FDA-approved for a multitude of cancers and transformed the treatment landscape, but significant obstacles remain and limit their efficacy [165]. Immunotherapies can activate a broad range of immune cell types with disparate effects on tumors, trigger severe side effects, exhibit low response rates, and rarely cure patients [166]. The inability to predict responders to treatment, lack of clinical trial designs that optimize efficacy, drug resistance, and high treatment costs further limit success. Since downregulation of MHC class I molecules on tumor cells is an important mechanism of immune escape and acquired checkpoint inhibitor resistance, a novel gain-of-function strategy such as proteasome activation that upregulates the antigen presentation machinery and increases MHC class I expression may improve or restore antitumor cellular immunity for clinical benefit. Importantly, the results also suggest that proteasome activators that amplify and unmask the tumor immunopeptidome may dictate the efficacy of T-cell immunotherapy, in contrast to other proposed mechanisms, e.g., PD-L1/PD-L2 expression, TMB, and microsatellite instability. Kalaora et al. demonstrated a correlation between elevated expression of PSMB8 and PSMB9 with heightened responsiveness to anti-CTLA4 and anti-PD1 therapies and improved survival rates in melanoma patients [167]. The findings suggest that the expression of immunoproteasome subunits may serve as biomarkers that predict response to immune checkpoint inhibitors and patient survival. Proteasome activators that generate and expand the presentation of TAAs, as well as public and private NeoAgs, may increase the antitumor efficacy of endogenous T-cells, genetically engineered CAR T-cells, and cancer vaccines. 

## Figures and Tables

**Figure 1 cancers-15-05632-f001:**
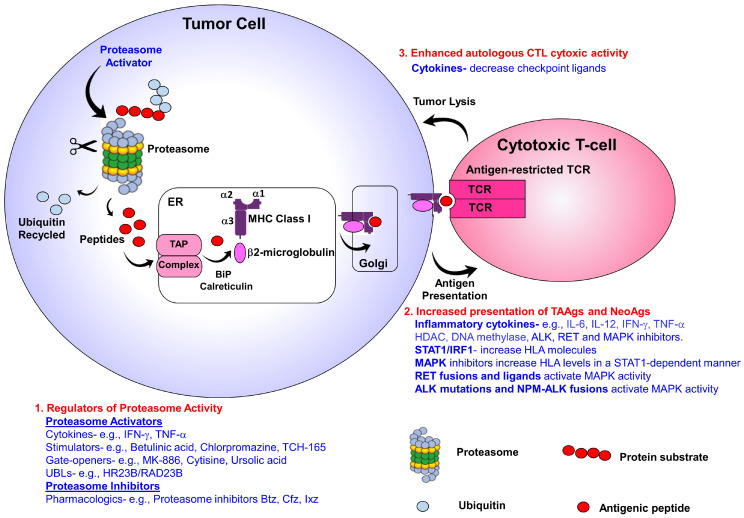
Shown is a graphical representation of the effect of proteasome activators on the degradation of intracellular UPS substrates, MHC-I antigen processing, transport, and presentation, and the effect on cytotoxic activity of TCR-restricted T-cells. Novel small molecules and FDA-approved pharmacologics provide the potential to activate proteasome activity and provide a feasible approach to expand and amplify the tumor immunopeptidomic landscape.

**Figure 2 cancers-15-05632-f002:**
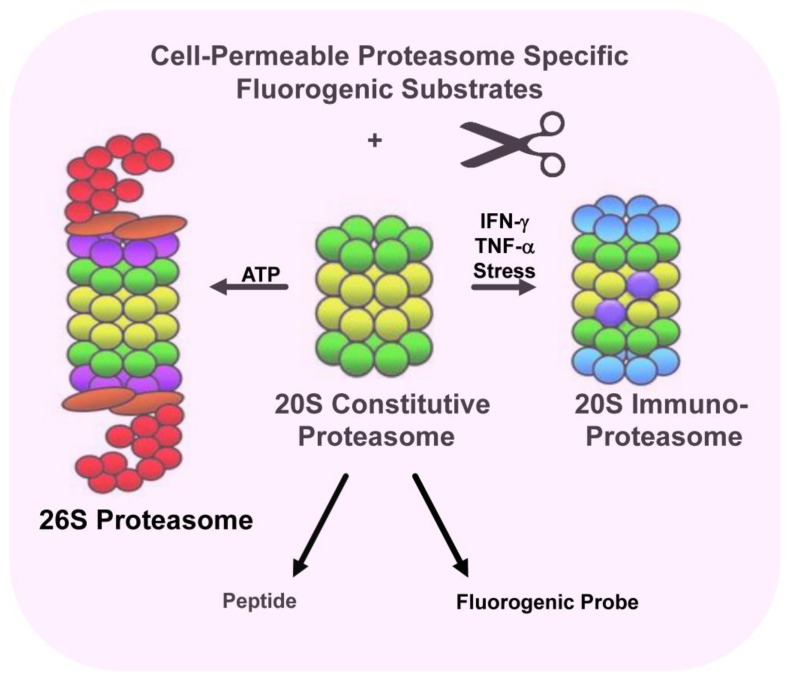
Model system using fluorogenic peptide substrates to screen compound libraries for proteasome modulators.

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
