# Peer review of "Targeting Proteasomes and the MHC Class I Antigen Presentation Machinery to Treat Cancer, Infections and Age-Related Diseases"

_cancers, 2023, doi:10.3390/cancers15235632_

Round 1

Reviewer 1 Report

Comments and Suggestions for Authors

The publication is a very comprehensive and detailed review of the current knowledge on the role of the proteasome as a potential therapeutic target. The authors draw attention to the proteasome as not only an important potential therapeutic target in anticancer therapy, but also its role in the development of neurodegenerative diseases or in the course of infections. The are also accurate suggestions in the summary regarding the proposed direction of development of therapeutic strategies are also valuable.

Author Response

We thank Reviewer #1 for the insightful comments. 

Reviewer 2 Report

Comments and Suggestions for Authors

The topic of this review article is a proteasome as a therapeutic target for treatments of cancer, infectious diseases, neurodegenerative diseases and aging. In the first two, introductory chapters, a basic information about proteasomes and immunoproteasomes is provided. Chapters II. - V. are not well-focused  since they deal rather with antigen presenting mechanisms in general than with proteasome targeting. On the other hand, chapter VI, dedicated to the role and targeting of proteasomes in neurodegenerative disease is well-written and provides useful information. The topic Proteasomes and aging deserves more space. Notably, a comprehensive review on a similar topic has been published recently in Cancers (Tundo GR et. al, Cancers 2021, 13, 4852). Collectively, this manuscript  should be substantially re-written, better focused and completed before publication.

Specific comments:

-          In chapters III.-V.,  there is not much about proteasome targeting, it is rather about  antigen presentation in general. Mostly antigen presentation defects including the MHC class I downregulation, and modulation in respect to the immune responses activation are discussed.

-          In Fig. 1, potential therapeutic targets are marked. Can you add a list of examples of  possible agents (cytokines, small molecules, etc.)?

-          It is not clear from you text whether ALK and RET inhibitors target proteasome or increase MHC class I.

-          You mention that the high-throughput screening of compound libraries may identify modulators of proteasome and immunoproteasome activity (Fig. 2). Can you specify the activators already discovered?

-          Particular proteasome inhibitors with a potential to treat infectious diseases  (chapter V) should be listed and discussed.

-          Lysosomatropic agents, such as chloroquine, have been described to influence also proteasomes. Can you  summarize this point?

-          Proteasome inhibitors, namely Bortezomib, been used for cancer therapy. Interestingly, their combination with immune response and proteasome activators (e.g. STING agonists) may be synergistic. Can you  discuss this point and, in general, the possibilities of the therapeutic use of proteasome modifiers combined with immunotherapeutic modalities?

Round 2

Reviewer 2 Report

Comments and Suggestions for Authors

The revised  version of the manuscript has been substantially modified, completed and the reviewers’ comments were reasonably addressed. This review article represents a comprehensive summary of the topic that will be useful  for experts in the field.

Minor point:

There are some typos in the added text.

Comments on the Quality of English Language

There are some typos in the added text (e.g. in the Title there should probably be  “Age-related Diseases” instead of “Age-related Disease”; please check the last sentence of the changed text in Chapter III, page 6 top).